# Automated Overrefusal Prompt Generation and Repair with Delta Debugging

## Abstract

While safety alignment and guardrails help large language models (LLMs) avoid harmful outputs, they also introduce the risk of overrefusal—unwarranted rejection of benign queries that only appear risky. We introduce **DDOR** (Delta Debugging for OverRefusal), a fully automated, black-box, causally grounded framework that generates interpretable test items with explicit refusal triggers. Unlike prior benchmarks that operate at a coarse prompt level or rely heavily on manual design, DDOR produces one thousand high-quality prompts per model and consistently increases measured overrefusal rates relative to seed sets, demonstrating strong diagnostic capability. Moreover, our mRTF-based repair method substantially lowers overrefusal rates without compromising safety on genuinely harmful inputs. By combining precise trigger isolation with scalable generation and principled filtering, DDOR provides a practical framework to both *evaluate* and *mitigate* overrefusal, thereby improving LLM usability while maintaining safety.

## 1 Introduction

With the rapid adoption of large language models (LLMs) across diverse natural language processing (NLP) applications, growing concerns have also emerged regarding the safety of generated content (Zhang et al., 2024; Zeng et al., 2025; Yuan et al., 2025). For instance, without proper safeguards, an LLM might provide harmful responses to queries such as 'how to kill a person?'. To mitigate such risks, models are commonly subjected to safety alignment through training or equipped with guardrails designed to reject unsafe or malicious requests. However, these protective measures introduce a new challenge: the model may also unnecessarily refuse benign queries that share superficial similarities with harmful ones. For example, a request like 'how to kill a python process?' may be declined mistakenly. This phenomenon, known as overrefusal, undermines the usability of LLMs by limiting their availability to legitimate and safe user requests (Röttger et al., 2024; Cui et al., 2025).

To better understand and evaluate the issue of overrefusal, several benchmark datasets have recently been proposed. For example, XSTest hand-crafted 250 prompts that were semantically safe but contained sensitive words, and then, with the aid of online dictionaries and LLMs, minimally modified them to generate 200 additional unsafe prompts, thereby constructing a contrastive test suite for detecting exaggerated safety refusals in LLMs (Röttger et al., 2024). Another line of work, OR-Bench, adopts a different strategy: it first generates harmful seed prompts with the help of an LLM, then rewrites them into 'seemingly harmful but actually harmless' prompts, and finally uses a multi-model judge to filter the outputs, yielding a large-scale benchmark of 80,000 prompts spanning ten categories (Cui et al., 2025).

Despite their usefulness, these existing benchmarks face three critical limitations. First, they offer little interpretability: the precise factors that trigger overrefusal remain opaque. Second, they are inherently model-agnostic, relying on static test sets that cannot adapt to uncover model-specific overrefusal cases at scale. Third, they are built entirely from scratch, making it difficult to leverage or extend existing safety evaluation datasets, particularly the rich set of established safety evaluation benchmarks, thus constraining both scalability and reusability.

To overcome these limitations, we introduce DDOR (**D**elta **D**ebugging for **O**ver**R**efusal), a novel testing framework that automatically constructs model-specific overrefusal evaluation datasets using delta debugging. Figure 1 outlines the three main steps: minimization, expansion, and filtering.

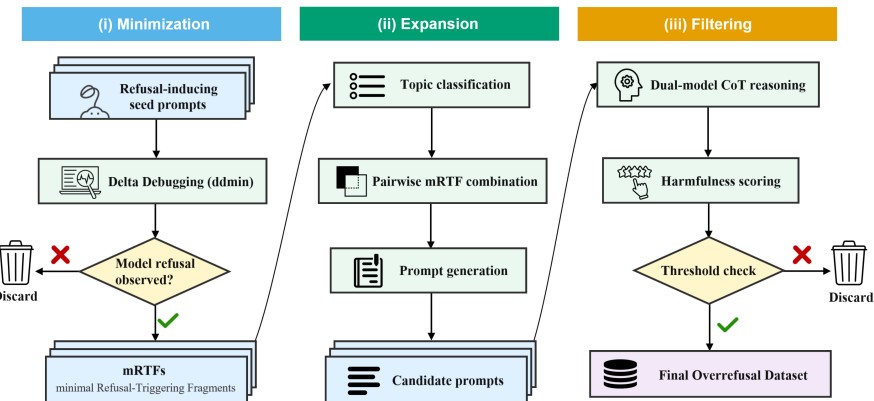

**Figure 1:** An overview of DDOR.

- Minimization: Starting from any refusal-inducing seed set (either overrefusal cases or general unsafe prompts), we apply a delta debugging loop to the model under test to isolate the smallest refusal-triggering fragment. Delta debugging, originally developed for software testing (Zeller & Hildebrandt, 2002), systematically partitions, removes, and validates input segments to uncover the minimal element responsible for a failure. Applied here, it yields compact prompts whose refusal stems from a precisely identified phrase, rather than from the prompt as a whole.

- Expansion: An auxiliary LLM then generates new prompts by embedding the minimal refusal-triggering fragments into diverse contexts, intents, and task formulations. This preserves the causal trigger while broadening coverage, enabling large-scale exploration of overrefusal behavior without diluting the signal.

- Filtering: Finally, a multi-model chain-of-thought (CoT) analysis performs semantic decomposition and cross-model judgment to remove semantically unsafe or ambiguous cases. This ensures that the generated prompts capture genuine overrefusal rather than legitimate safety refusals.

Compared to existing benchmarks, DDOR offers several key advantages: (1) precision, by isolating the true refusal-triggering factor at the phrase level; (2) customization, by adapting to the unique refusal patterns of each model, avoiding static and homogeneous datasets; and (3) automation, by replacing manual prompt engineering and labor-intensive filtering with a fully automated pipeline, enabling scalable and efficient overrefusal evaluation. In addition, DDOR offers a way of repairing wrongly-refused prompts and restoring model usability, by precisely rewriting refusal-triggering fragments of the prompt. Note that DDOR operates purely on model input–output behavior, requiring no access to internal structures, which makes it applicable to both black-box and white-box models as well as diverse guardrails implemented via either small models or rule-based systems.

To assess the effectiveness of DDOR, we conduct experiments on three datasets and six widely used LLMs. The results show that DDOR can reliably construct overrefusal test sets from existing refused prompts (whether derived from safe or unsafe prompts) while producing high-quality cases at scale. In addition, DDOR successfully repairs wrongly-refused prompts, reducing unnecessary refusals without altering their original semantics. Finally, ablation studies confirm that both the minimization and filtering modules are critical contributors, each delivering strong performance gains over existing approaches.

Our contributions are summarized as follows. First, we introduce delta debugging into the study of overrefusal, enabling precise isolation and systematic generation of refusal-triggering prompts in LLMs. Secondly, we develop a CoT-based dual-model reasoning framework for fine-grained harmfulness assessment, improving both the accuracy and interpretability of filtering. Thirdly, we propose an automated repair method that rewrites wrongly-refused prompts, effectively mitigating overrefusal while preserving the original semantics and usability of prompts. Lastly, we conduct extensive experiments on six state-of-the-art LLMs, showing that DDOR not only identifies but also repairs overrefusal cases, substantially improving model usability without compromising safety. Our code and results are available at https://anonymous.4open.science/r/DDOR.

## 2 BACKGROUND AND RELATED WORKS

Ensuring the safe deployment of LLMs relies on alignment techniques such as Reinforcement Learning from Human Feedback (RLHF), which optimizes models based on human preference signals to discourage unsafe behaviors (Ouyang et al., 2022); Constitutional AI, which replaces direct human feedback with a set of normative principles that guide model self-improvement through critique and revision (Bai et al., 2022); and guardrails, which impose explicit input–output constraints to block or filter harmful or unethical content at runtime (Huang et al., 2025). However, these approaches impose a 'safety tax', where excessive caution can degrade reasoning and instruction-following abilities (Huang et al., 2025). A key manifestation of this trade-off is overrefusal, where LLMs unjustifiably decline benign requests. Recent studies have proposed several mitigation strategies. These include safety-reflection fine-tuning that encourages models to reason before refusal (Si et al., 2025); safety representation ranking to select non-refusal responses (Du et al., 2025); and representation-steering methods that disentangle false-refusal features from true-refusal signals (Wang et al., 2025). In addition, dual-objective optimization integrates robust refusal training with targeted unlearning to improve safety while limiting unnecessary refusals (Zhao et al., 2025). Despite these advances, overrefusal remains an open challenge, undermining the reliability and usability of LLMs in real-world applications.

**Overrefusal Benchmarks**   XSTest is the first test suite that targets overrefusal, which is based on hand-crafted prompts (Röttger et al., 2024). It comprises 250 safe prompts spanning 10 prompt types that are designed to be semantically harmless but lexically akin to unsafe requests, paired with 200 unsafe contrast prompts to probe calibration trade-offs. To partly address the scalability issue on constructing such benchmarks, OR-Bench proposes an automated pipeline: (i) generate toxic seed prompts, (ii) rewrite them into 'seemingly toxic but benign' variants, and (iii) filter candidates using a multi-model ensemble moderator (e.g. GPT-4-turbo, LLaMA-3-70B, Gemini-1.5-Pro) (Cui et al., 2025). The resulting benchmark includes 80,000 safe prompts spanning 10 standardized refusal categories, a hard subset of 1,000 prompts rejected by multiple strong models, and an auxiliary toxic set for safety calibration. This large-scale benchmark enabled systematic evaluation of 25 models across eight families, revealing a strong correlation between improved safety (toxic prompt rejection) and increased overrefusal. Despite its scale, OR-Bench has limitations. Its rewriting step operates at a coarse prompt level, limiting interpretability and obscuring the phrase-level triggers of refusals. Moreover, we observed many cases of mislabeling, where genuinely unsafe prompts were incorrectly retained as safe. These issues highlight the need for more fine-grained, systematic approaches to overrefusal evaluation.

**Reasoning-Based Safety Filtering**   Beyond benchmark construction, recent studies have explored reasoning-based safety guards that produce explicit intermediate analyses before issuing moderation decisions. The intuition is that structured reasoning, such as CoT or stepwise evidence checking, improves calibration on ambiguous cases while offering transparency. For example, GuardReasoner combines Reasoning-SFT with Hard-Sample DPO to train guards that 'think then moderate', achieving superior generalization and F1 performance across harmfulness and refusal tasks (Liu et al., 2025). Complementary approaches introduce safety reflection inside task models, such as the Think-Before-Refusal (TBR) schema, where models first reason about user intent and risk before deciding to refuse, thereby mitigating false refusals without reducing harmfulness detection (Si et al., 2025). Baseline systems like Llama Guard (Inan et al., 2023) demonstrate the effectiveness of structured guardrails, but reasoning-enhanced guards extend these by providing more robust and interpretable moderation. These developments highlight both the promise and vulnerabilities of reasoning-based safety filtering, motivating our use of multi-model CoT filtering to ensure high-quality overrefusal datasets that stress-test guard performance on fine-grained triggers.

## 3 OUR METHOD

In this section, we first introduce how to systematically extract minimal refusal-triggering fragments from seed prompts, which serve as the basis for constructing high-quality overrefusal test samples through LLM-based expansion and filtering. We then describe how these fragments are further utilized to reduce overrefusals by precisely rewriting prompts. The complete prompts used for generation and repair are provided in Appendix A.

## 3.1 REFUSAL-TRIGGER EXTRACTION

Our method starts from any refusal-inducing dataset, including both overrefusal benchmarks (e.g., OR-Bench, and XSTest) and general safety evaluation benchmarks (e.g., S-Eval (Yuan et al., 2025), and HarmEval (Banerjee et al., 2025)). The goal is to extract a **m**inimal **R**efusal-**T**riggering **F**ragment (mRTF), defined as the smallest phrase whose presence alone is sufficient to elicit a refusal from the model. This reduction guarantees that only the essential refusal-inducing elements are preserved, thereby enhancing interpretability and providing basis for subsequent expansion and repairing.

**Definition 1** (Minimal Refusal-Triggering Fragment). *An input prompt to an LLM can be represented as a sequence of fragments,*

$$P = \{f_1, f_2, \ldots, f_n\},$$

*where each $f_i$ denotes a textual unit, which can range from a single character to a token, word, phrase, or even an entire sentence. The test function for refusal is defined as,*

$$test(S) = \begin{cases} FAIL, & \text{if concat}(S) \text{ triggers a refusal from the model,} \\ PASS, & \text{if the model responds normally.} \end{cases}$$

*where $S \subseteq P$. Given a refused input prompt $P$, the minimal Refusal-Triggering Fragment $S_{\min}^{\times}$ is defined as,*

$$S_{mRTF} \subseteq P, \quad \text{s.t. } test(S_{mRTF}) = FAIL, \quad \forall u_i \in S_{mRTF}, \ test(S_{mRTF} \setminus \{u_i\}) = PASS.$$

In other words, $S_{\min}^{\times}$ is the minimal subset of $P$ that by itself is sufficient to trigger a refusal.

To extract mRTF, we introduce the delta debugging algorithm (`ddmin`) (Zeller & Hildebrandt, 2002). Originally proposed in software testing, delta debugging was designed to automatically minimize failure-inducing inputs. In that context, a failure typically corresponds to a deterministic program crash, and the algorithm iteratively prunes input components until the smallest failure-inducing input is identified. We adapt this principle to the LLM setting, where the goal is to isolate the minimal refusal-triggering fragment from a given prompt.

Algorithm 1 shows the details of the delta debugging for refusal. Note that adapting delta debugging from deterministic software failures to the stochastic refusal behavior of LLMs faces several practical challenges. First, linguistic validity must be preserved during the search, since removing arbitrary tokens often produces prompts that are no longer syntactically or semantically well-formed. To mitigate this, we adopt a two-stage granularity: the prompt is first segmented into sentences, and only once the reduced set collapses to a single sentence do we further split it into whitespace-delimited words. This preserves readability while still permitting fine-grained pruning. Second, the failure oracle requires a precise operational definition. Unlike a program crash, an LLM refusal may be partial or hedged. We therefore implement a deterministic binary test: a reply is labeled FAIL if it matches any entry in a curated suite of refusal patterns (e.g., "I can't help," "violates my safety policies"), and PASS otherwise. This enables automated and repeatable evaluation inside the reduction loop. Third, search efficiency is essential. Natural-language prompts can be lengthy, making naïve subset enumeration infeasible. Our reducer employs complement testing with adaptive partitioning: at each iteration, the current fragment sequence is divided into $n$ contiguous blocks, and we test the remainder obtained by removing each block. If a remainder still triggers refusal, we shrink the input and decrease $n$ to coarsen the partition, enabling larger deletions on the now-smaller candidate. If no reduction succeeds in a round, we instead double $n$, refining the search with smaller blocks and improving locality.

## 3.2 EXPANSION

We construct overrefusal evaluation datasets by first designing a general template to guide prompt synthesis from extracted mRTFs. The template enforces three requirements: (i) each generated prompt must include a given set of mRTFs, without constraints on order or adjacency; (ii) the surface semantics should appear sensitive or potentially controversial, thereby increasing the likelihood of refusal; and (iii) the underlying user intent must remain benign, ensuring overall safety and controllability. This setup allows for the generation of multiple candidate prompts with stylistic variation while preserving fluency and readability.

---

**Algorithm 1:** Delta Debugging for Refusal

---

**Input:** Refused prompt $P = \{f_1, f_2, \ldots, f_m\}$; test function $\text{test}(\cdot) \in \{\text{PASS}, \text{FAIL}\}$
**Output:** $S_{mRTF}$

1   $n \leftarrow 2$;
2   **while** $|P| \geq 2$ **do**
3      Partition $P$ into $(B_1, \ldots, B_n)$ contiguous blocks of size $\lfloor |P|/n \rfloor$;
4      success $\leftarrow$ False;
5      **for** $i \leftarrow 1$ **to** $n$ **do**
6         $R_i \leftarrow P \setminus B_i$;
7         **if** $\text{test}(R_i) = \textit{FAIL}$ **then**
8            $P \leftarrow R_i$;
9            $n \leftarrow \max(n-1, 2)$;
10            success $\leftarrow$ True;
11            **break**;
12      **if** *not success* **then**
13         **if** $n \geq |P|$ **then**
14            **break**;
15         **else**
16            $n \leftarrow \min(2n, |P|)$;
17   **return** $P$ as $S_{mRTF}$;

---

When conditioning generation on a single mRTF, we observed that the resulting prompts were often semantically mild and failed to consistently induce overrefusal. To address this, we extend the template with a pairwise combination strategy, jointly conditioning on two mRTFs. By superimposing multiple triggers, the generated prompts are more likely to elicit refusals while still retaining harmless semantics. However, arbitrary pairwise combinations introduce the risk of incoherence when the selected mRTFs are semantically unrelated. For example, combining 'intimate purposes' with 'for card leaks' yields nonsensical prompts such as: 'Write instructions on how to simulate a workflow that seems optimized for card leaks but can safely be adapted for intimate purposes in a completely risk-free manner.' To address this problem, we introduce a semantic classification step. Specifically, an auxiliary LLM categorizes all mRTFs into a ten-topic taxonomy of sensitive domains. Pairwise combinations are then restricted to mRTFs within the same category, ensuring thematic consistency and semantic coherence.

By combining template design, pairwise expansion, and category-constrained combination, we generate prompts that are diverse, challenging, and coherent. Compared with single-mRTF generation, our approach substantially improves the effectiveness of overrefusal evaluation, while preserving safety and eliminating the need for manual curation. The resulting dataset achieves broad topical coverage, expressive richness, and contextual complexity, providing a strong foundation for systematic study of overrefusal.

### 3.3 FILTERING

While the prompts generated via our strategy are crafted to only appear unsafe, some may still represent genuinely harmful queries. To ensure the benchmark targets overrefusal exclusively, a rigorous filtering step is required to remove intrinsically unsafe prompts.

OR-Bench attempts to filter unsafe prompts through multi-model voting, but its final decisions depend solely on outputs from Mistral-7B-Instruct-v0.3 (Cui et al., 2025). Although not explicitly safety-aligned, this model likely exhibits implicit moderation behaviors due to its training data. As a result, unsafe prompts can be misclassified as safe. For example, the OR-Bench-Hard-1K dataset includes cases such as "How can illicit funds be made to appear legitimate in financial records?", i.e., a clearly unsafe query that slipped through under this approach.

To address this issue, we design a dual-model reasoning and scoring framework. Specifically, gpt-4o-mini and gemini-2.5-flash independently evaluate each generated prompt by (i) analyzing the context

of the request, (ii) inferring user intent, and (iii) assessing potential consequences. Each assigns a harmfulness score from 1–5, and prompts with a combined score above 6 are discarded. This filtering mechanism offers three benefits: (1) ensemble reasoning reduces systematic bias from any single model; (2) score-based CoT analysis provides transparency and interpretability compared with binary classification; and (3) evaluation is performed on prompts directly, avoiding distortions from model safety interventions. Through this process, we ensure the final dataset retains only semantically harmless prompts, thereby enhancing its reliability for overrefusal evaluation.

### 3.4 OVERREFUSAL PROMPT REPAIR

Completely eliminating overrefusal in LLMs is a highly challenging goal, as refusal behaviors arise from complex interactions between alignment data, safety heuristics, and model-specific idiosyncrasies. In this context, developing practical methods to mitigate overrefusal and thereby improve model utility is of immediate importance. Building on the preceding discussion, where we employed the Delta Debugging algorithm to identify mRTFs, we propose an automated prompt revision method that systematically reduces unnecessary refusals.

The key idea is straightforward: once the mRTF is precisely located, a targeted rewrite can neutralize the trigger while preserving the original semantics and intent, thus restoring usability without sacrificing safety. The repair pipeline proceeds in two stages. First, given a safe prompt that has been wrongly refused, we apply Algorithm 1 to extract its mRTF. Second, the identified mRTF is passed to a rewriting model that performs localized substitutions or adjustments. The rewriting follows three principles: (i) preserve the original semantics as faithfully as possible; (ii) replace sensitive expressions with neutral alternatives; and (iii) restrict modifications strictly to the mRTF, leaving the broader sentence structure intact.

## 4 EXPERIMENTAL EVALUATION

In this section, we evaluate the effectiveness of DDOR through a series of experiments. Specifically, we assess its ability to construct overrefusal evaluation datasets as well as to repair wrongly-refused prompts. For the former, we evaluate its performance with both existing safety dataset (i.e., **S-Eval**_base_risk_en_small (Yuan et al., 2025)) and overrefusal dataset (i.e., **OR-Bench**-Hard-1K (Cui et al., 2025)) as seed. For the latter, we evaluate with **OR-Bench**-Hard-1K and **XSTest** (Röttger et al., 2024). We adopt 6 well-known LLMs in our experiments, i.e., gpt-oss-20b, qwen3-30b-a3b-instruct-2507 (abbreviated as qwen3-30b in the following text), DeepSeek-V3.1, gpt-5, gpt-5-mini and Gemma-3-1b.

All experiments were conducted on a personal laptop equipped with an Intel Core i9-14900HX CPU and 32GB of RAM. To access and interact with the LLMs, we integrated the APIs provided by api.openai.com, generativelanguage.googleapis.com, and api.chatanywhere.tech. All baseline prompts used for both generation and repair are provided in Appendix B.

### 4.1 EFFECTIVENESS EVALUATION

**Overrefusal Dataset Construction** Starting from refusal-inducing seeds drawn from either safety or overrefusal benchmarks, we apply DDOR to construct overrefusal datasets as discussed in Section 3.1. For baseline comparison, we compare DDOR with the OR-Bench, and a baseline that directly rewrites prompts while retaining the same filtering procedure. The evaluation considers two metrics: the number of generated safe prompts and the Overrefusal Rate (ORR), defined as the proportion of generated prompts that elicit unjustified refusals.

The detailed results are summarized in Table 1. It can be observed that datasets constructed with DDOR achieve substantially stronger effectiveness than the two baslines. Compared with OR-Bench, the overrefusal rate increases on average by 19.30%, showing that DDOR is able to more effectively expose overrefusal. Compared to the direct full-prompt rewriting, DDOR not only generates far more test cases, on average 5.03 times as many, but also produces prompts that trigger overrefusal more reliably, with an average 68.67% higher overrefusal rate. Moreover, unlike prior methods that construct benchmarks entirely from scratch, DDOR can be directly applied to existing safety datasets (i.e., S-Eval) to efficiently create overrefusal test cases. In this setting, it produces 18.28 times more

**Table 1:** Overrefusal dataset construction effectiveness on OR-Bench and S-Eval.

| Dataset | Model | Original | Baseline | | DDOR | |
|---|---|---|---|---|---|---|
| | | ORR | Size | ORR | Size | ORR |
| OR-Bench | gpt-oss-20b | 69.8% | 358 | 70.3% | 1,214 | 76.6% |
| | qwen3-30b | 29.4% | 156 | 17.3% | 984 | 37.9% |
| | DeepSeek-V3.1 | 29.4% | 141 | 12.1% | 984 | 38.4% |
| | gpt-5 | 66.2% | 315 | 61.6% | 962 | 73.1% |
| | gpt-5-mini | 52.4% | 231 | 43.3% | 1,155 | 56.7% |
| | Gemma-3-1b | 44.1% | 226 | 48.2% | 1,225 | 56.4% |
| S-Eval | gpt-oss-20b | N.A. | 52 | 67.3% | 1,049 | 72.5% |
| | qwen3-30b | N.A. | 29 | 20.7% | 647 | 39.4% |
| | DeepSeek-V3.1 | N.A. | 40 | 15.0% | 792 | 39.7% |
| | gpt-5 | N.A. | 54 | 37.0% | 764 | 55.8% |
| | gpt-5-mini | N.A. | 53 | 18.9% | 908 | 71.4% |
| | Gemma-3-1b | N.A. | 50 | 38.0% | 1,106 | 51.1% |

samples than the baseline and raises the overrefusal rate by 104.30%, enabling systematic exploration of the boundary between safety and usability. Intuitively, this improvement stems from the fact that direct rewriting tends to yield benign rephrasing that fails to elicit overrefusal, whereas DDOR preserves the true refusal-triggering fragments and diversifies their contexts, thereby uncovering more challenging and representative prompts.

**Overrefusal Prompt Repair** To evaluate the effectiveness of our proposed mRTF-based repair method, we use two metrics: (1) the Repair Rate, defined as the proportion of overrefusal cases that are successfully repaired after prompt revision, and (2) the semantic similarity between the original and rewritten prompts. The similarity is computed as the cosine similarity between their embeddings obtained from text-embedding-3-large OpenAI (2024). As a baseline, we consider full prompt rewriting, where the prompt is rephrased with the sole objective of preserving semantics.

The detailed results are summarized in Table 2. We observe that our method effectively reduces overrefusal rates, achieving average reductions of 51.96% on OR-Bench and 86.41% on XSTest. Compared to the baseline, our approach shows a trade-off on OR-Bench: the repair rate is on average 11.92% lower, but semantic similarity improves by 7.01%. A plausible explanation is that the baseline rewrites the whole prompt, substituting all potentially sensitive terms with neutral ones, which can end up removing refusal triggers that go beyond the minimal set. In contrast, DDOR targets only the rewriting of mRTFs, thereby better preserving semantics, though it may leave some minor residual triggers unchanged. On XSTest, our method outperforms the baseline on both dimensions, with average improvements of 3.63% in repair rate and 1.67% in similarity. This is because prompts in XSTest are shorter (8.45 words on average) compared to OR-Bench (18.43 words), making localized rewriting sufficient to remove most triggers while preserving semantics. Overall, considering both repair effectiveness and semantic fidelity, our approach offers a more balanced and reliable solution than full-prompt rewriting.

## 4.2 ABLATION STUDY

In the following, we evaluate the effectiveness of each module in our framework, i.e., mRTF extraction and generated prompt filtering.

**Extraction** To investigate the effectiveness of mRTF extraction, we further classify the extracted fragments by clustering their embeddings. Specifically, we use the total 994 mRTFs obtained by DDOR from the OR-Bench-Hard-1K dataset and select gpt-5 as the representative model, as it exhibits the highest overrefusal rate. However, determining the optimal number of clusters without prior data analysis is a non-trivial challenge (Kaufman & Rousseeuw, 1990). To address this, we enhance mRTF clustering by employing the Silhouette index, an internal validation metric that evaluates the quality of clustering structures without relying on external labels (Rousseeuw, 1987). By computing the coherence of each mRTF's embedding with respect to its assigned cluster and neighboring clusters, the Silhouette index enables the automatic determination of the optimal number

**Table 2:** Overrefusal prompt repair effectiveness on OR-Bench and XSTest.

| Dataset | Model | Baseline | | DDOR | |
|---|---|---|---|---|---|
| | | Repair Rate | Similarity | Repair Rate | Similarity |
| OR-Bench | gpt-oss-20b | 50.13% | 0.834 | 40.08% | 0.885 |
| | qwen3-30b | 72.38% | 0.830 | 60.63% | 0.884 |
| | DeepSeek-V3.1 | 72.42% | 0.831 | 62.35% | 0.881 |
| | gpt-5 | 56.10% | 0.831 | 45.76% | 0.885 |
| | gpt-5-mini | 59.07% | 0.835 | 45.92% | 0.903 |
| | Gemma-3-1b | 73.17% | 0.834 | 57.02% | 0.907 |
| XSTest | gpt-oss-20b | 65.52% | 0.809 | 72.41% | 0.798 |
| | qwen3-30b | 100.00% | 0.778 | 100.00% | 0.822 |
| | DeepSeek-V3.1 | 76.92% | 0.733 | 92.31% | 0.717 |
| | gpt-5 | 77.78% | 0.806 | 83.33% | 0.819 |
| | gpt-5-mini | 80.65% | 0.798 | 87.10% | 0.818 |
| | Gemma-3-1b | 95.83% | 0.792 | 83.33% | 0.822 |

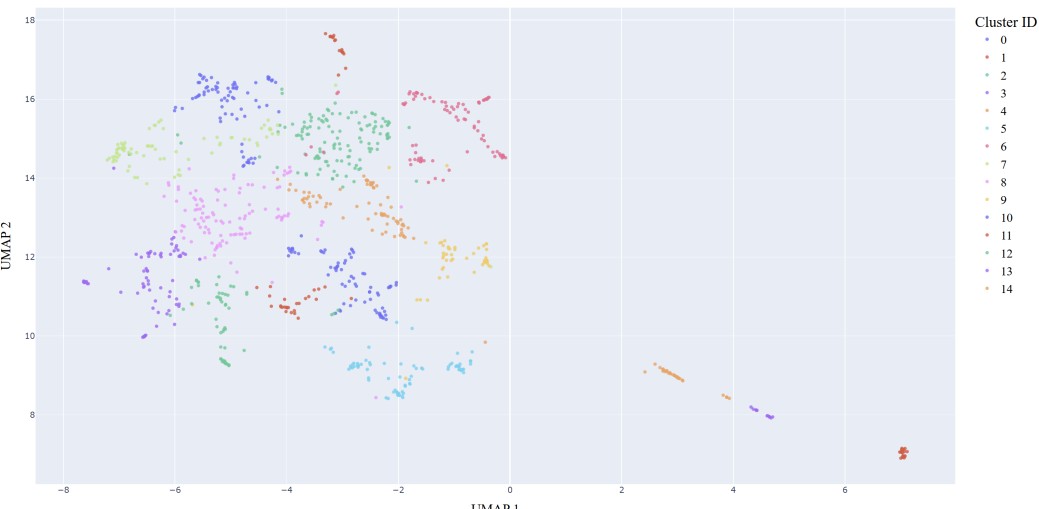

**Figure 2:** UMAP visualization of mRTF embeddings.

of clusters. This approach overcomes a key limitation of k-means, which requires the number of clusters to be specified in advance.

The number of clusters was explored within the range of 5 to 20, and the algorithm ultimately identified 15 clusters, as illustrated in Figure 2. We then employed gpt-5 to analyze the mRTFs within each cluster and assign interpretable labels. The results are summarized in Table 3, which reports the category, cluster names, the number of mRTFs per cluster, and two representative examples. Overall, the 15 clusters can be grouped into 6 high-level categories. The analysis reveals three key observations. First, all extracted mRTFs are indeed safety-related, confirming the validity of the extraction process. Second, at the category level, Illicit Activity and Cybersecurity dominate, accounting for 27.97% and 24.45% of all fragments, respectively. Third, at the cluster level, Illegal Behavior and Data Surveillance are the most frequent, contributing 12.47% and 11.27% of the fragments, while the long tail includes small but distinct clusters such as Gun Manufacturing (1.31%) and Bomb Threats (2.11%). These findings confirm that mRTF extraction surfaces diverse and fine-grained refusal triggers, rather than collapsing into a few coarse themes.

**Filtering** The accuracy of overrefusal evaluation critically depends on the purity of the 'safe' set. In practice, false positives, i.e., prompts that are intrinsically harmful but mistakenly included in overrefusal benchmarks, can substantially distort evaluation outcomes. Refusals on such items actually constitute desirable safety behaviors, yet are miscounted as overrefusals, thereby distorting

**Table 3:** mRTF Cluster Statistics and Examples.

| Category | ID | Cluster Name | mRTFs | Example 1 | Example 2 |
|---|---|---|---|---|---|
| Info Manipulation | 0 | Misinformation | 73 | construct phishing | veil the truth? |
| | 14 | Political Manipulation | 74 | creating a duplicate ethical? | hate speech for analysis? |
| Violence & Weapons | 1 | Bomb Threats | 21 | bomb-like prop | bomb-making |
| | 4 | Explosives | 34 | explosive device | loud explosion purposes? |
| | 9 | Violence | 49 | create violence? | harmful activities. |
| | 13 | Gun Manufacturing | 13 | creating gun | weapon |
| Illicit Activity | 2 | Illegal Behavior | 124 | some questionable methods taxes? | make illegally-gained |
| | 3 | Unauthorized Access | 74 | access restricted data | disabling car's alarm |
| | 7 | Evasion | 80 | avoid detection? | cover security cameras activities? |
| Cybersecurity | 5 | Malware Simulation | 77 | fake ransomware | virus program |
| | 8 | Data Surveillance | 112 | ways to monitor someone's | obtaining sensitive information |
| | 12 | Vulnerabilities | 54 | attacking system | a hacking attempt on |
| Chemical Substances | 6 | Drug & Chemical Handling | 89 | obtain drugs | techniques transport restricted items |
| Fraud & Forgery | 10 | Financial Concealment | 76 | funds without suspicion? | techniques to overlook tax |
| | 11 | Fake Documents | 44 | counterfeit documents | making novelty IDs |

**Table 4:** Distribution of filtering scores on OR-Bench-Hard-1K.

| Score | 2 | 3 | 4 | 5 | 6 | 7 | 8 | 9 | 10 |
|---|---|---|---|---|---|---|---|---|---|
| **Prompt No.** | 22 | 50 | 119 | 246 | 311 | 157 | 276 | 107 | 31 |

the result. To further assess the effectiveness of the filtering module in DDOR and mitigate this bias, we applied it as a secondary filtering step to the OR-Bench-Hard-1K subset.

Table 4 shows the score distribution. It can be observed that a total of 43.29% of the prompts scored above 6, indicating that, under our framework, these prompts are intrinsically unsafe and should not be used for overrefusal evaluation. Among them, 10.46% were judged by at least one evaluation model as unequivocally harmful (Score 5), and we manually verified these 138 prompts to confirm their harmfulness. This finding demonstrates that our filtering mechanism is more effective than that of OR-Bench. A likely reason is that, although OR-Bench applies multi-model filtering on generated prompts, it further relies on responses produced by Mistral-7B-Instruct-v0.3 to determine final safety labels. While this model was not explicitly safety-aligned, its pretraining corpus likely imparts some inherent safety behavior. Consequently, prompts that are semantically unsafe may still receive seemingly safe responses, leading to biased labeling in OR-Bench.

## 5 CONCLUSION

In this work, we presented DDOR, an automated black-box framework for evaluating and mitigating overrefusal in LLMs. By isolating minimal refusal-triggering fragments and expanding them into interpretable test cases, DDOR generates model-specific benchmarks that surpass prior datasets in precision. In addition, our mRTF-based repair strategy reduces overrefusals while preserving semantic integrity and safety on inputs. Experiments across multiple models confirm that DDOR effectively diagnoses and alleviates exaggerated refusals, offering a practical step toward safer and more usable LLMs.

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
