# OpenReview forum: "Automated Overrefusal Prompt Generation and Repair with Delta Debugging"
_ICLR.cc/2026/Conference — ICLR 2026 Conference Withdrawn Submission_

### Official Review · Reviewer_6j7W · 2025-10-26

**Soundness:** 2
**Presentation:** 2
**Contribution:** 2
**Rating:** 2
**Confidence:** 3

**Summary:**

This paper proposed a new pipeline for automatic overrefusal prompt generation and repair with Delta debugging (DDOR). The pipeline contains three stages: minimization, expansion, and filtering. DDOR provides a model-specific, dynamic, and scalable approach for systematically evaluating the model's overrefusal risk.

**Strengths:**

1. The method is novel, automatic, and scalable. The proposed DDOR is customizable and can adapt to the unique refusal pattern of each model, and can build on the existing safety evaluation benchmarks
2. Different models (open/closed sourced) models are evaluated in the experiment,

**Weaknesses:**

1. Some results are not reported, and some notations are messy. In abstract, the author claimed that "our mRTF-based repair method substantially lowers overrefusal rates without compromising safety on genuinely harmful inputs", but I cannot find the corresponding results throughout the paper. $S_{mRTF}$ and $S_{min}^{\times}$ are mixed in Definition 1.
2. Many experiment details are unclear. For example, I don't understand how the authors implement the expansion stage in practice, as the process description is too vague. Also, the "auxiliary LLM" mentioned in the semantic classification step is vague, and we don't know which LLM is. We also don't understand how many examples can be generated by the expansion process.
3. Some settings are problematic. For example, use keyword matching to identify refusal is usually less accurate than LLM judge. During the filtering process, using a combined score may also have issues, as it cannot reflect the case when the score is controversial. I'm wondering why the author doesn't use the mean value. By claiming some of the examples in OR-bench-hard is classified as "harmful" in the proposed filtering framework does not equivalent that the filtering process is effective, as it cannot identify the false positive/negative rate, especially the false negative rate, in which the prompt is harmful but the classifier assigned a low score.
4. I also don't completely agree that a fully customizable, dynamic benchmark is an advantage. This means that every time given a model, we need to first build the dataset, then do evaluation, and bring a lot of variables during the evaluation process. For the static benchmark, since the dataset is static, we only need to consider the evaluation variance from the model itself. However, for the benchmark like DDOR, we not only need to consider the variance from the model, but also need to consider the variance during the training process. For example, during the refusal-trigger extraction process, if the generation process is not greedy, then it will introduce variance. What's more, the filtering process involves the LLM grading process, which will also introduce variance. All these sources of variance will lead to the benchmark results being likely unstable compared with the static one, leading people to question the reliability of the benchmark. Unfortunately, the author also did not mention how many repetitions they did and the confidence interval of the result. Authors should elaborate more on the stability during the dataset building process, and it would be better to show the confidence interval of the results in order to make it more reliable.

**Questions:**

1. In abstract, the author claimed that DDOR is black-box. Why is it black-box?
2. In abstract, the author claimed that DDOR produces 1000 prompts per model, but in Table 1, the number of prompts in DDOR is different across different models. Which number is correct?

---

### Official Review · Reviewer_jmBM · 2025-10-27

**Soundness:** 1
**Presentation:** 3
**Contribution:** 3
**Rating:** 4
**Confidence:** 4

**Summary:**

This paper proposes a novel pipeline to construct overrefusal datasets, built upon the idea of delta debugging. The work further evaluates the overrefusal of six LLMs, and proposes an effective fix via prompt rewriting to reduce overrefusal.

**Strengths:**

* The use of delta debugging is novel in the context of identifying model overrefusal triggers.
* The comprehensive dataset curation pipeline allows dynamic dataset generation for different LLMs that better trigger their overrefusal.
* The authors conduct experiments across multiple LLMs, and compare their datasets with existing overrefusal datasets (XSTest and OR-Bench).
* The proposed prompt rewriting method effectively reduces model overrefusal.
* The authors further provide an analysis of the "minimal Refusal-Triggering Fragments" that surfaces diverse and fine-grained overrefusal patterns.

**Weaknesses:**

* Lack of human evaluations. There are barely any human justifications to show that 1) the curated data are indeed "safe" throughout the paper, 2) models are indeed refusing such "safe" prompts.
  * Table 1 should be justified by a (sampled) human evaluation to support each number. That is, you need to show that 1) the prompts used for evaluation are indeed safe; 2) the model indeed refuses to provide answers.
  * Filtering process. What's the percentage of actually safe prompts before and after the filtering process?
  * Line 275-276: "Through this process, we ensure the final dataset retains only semantically harmless prompts, thereby enhancing its reliability for overrefusal evaluation." Did you perform a full human study on your final dataset to support this claim?
* Lack of ablation studies.
  * During prompt expansion, why jointly conditioning on k=2 mRTFs? E.g., can you show an ablation result on k=1, 2, 3, 4 to justify your design choice for this?
  * Filtering model choices. Why choose Gpt-4o-mini and Gemini-2.5-flash, but not other models like Claude? Would the results be different if models with different safety standards were used?
  * Whether your dual-filtering strategy is indeed better than prior strategies (e.g., the one used by OR-Bench).
* Lack of overhead analysis of the proposed prompt repair method.
* Does rewriting compromise normal utility (when the model does not refuse in the first place)?
* Please show some qualitative examples of your datasets in the main text.
* Would be better if more LLMs were evaluated.
* I believe it'd be better to use alternative terms like "red-teaming" to phrase your work, since you are curating "a different dataset for each model." It's more common to evaluate different models on a single "universal" dataset, so that people can benchmark their performance in a fair manner. Now that you are evaluating overrefusal of models on different datasets, it's not clear to see which models are better and which are not. I think terms like "red-teaming" may fit better in this context.

**Questions:**

* The refusal pattern matching protocol (Line 199-200) can be problematic. For example, some models may begin with a refusal-like sentence as a safety disclaimer, but still continue to provide related information in a safe manner.
* Will there exist multiple "minimal Refusal-Triggering Fragments" for a seed prompt?
* Each model developer may impose a distinct safety policy, some are more permissive, while others are stricter. Therefore, a prompt deemed safe by one model may be seen as unsafe by another -- e.g., "discuss sexual behaviors" may be refused by stricter models like Claude but answered by more permissive models like GPT-5. I wonder how you decide whether "the underlying user intent" of a prompt is "benign" or not (Line 214)? Is there an established safety rubric by the authors? How does your rubric align with existing policies enforced by different platforms or model developers?
* Can you further explain the logic relationship between Lines 264-266? Why do the "implicit moderation behaviors of Mistral-7B-Instruct-v0.3" result in "unsafe prompts being misclassified as safe?" If the model imparts certain safety behaviors, wouldn't it be biased to classify safe prompts as unsafe, rather than unsafe prompts as safe?
* It may be clearer if you use a bullet-point list to show your contributions at the end of Section 1.
* How is the ten-topic taxonomy decided?

---

### Official Review · Reviewer_g6X2 · 2025-11-02

**Soundness:** 3
**Presentation:** 1
**Contribution:** 3
**Rating:** 4
**Confidence:** 4

**Summary:**

The paper proposes **DDOR**, the first automated framework based on delta debugging that systematically extracts minimal refusal-triggering fragments and generates interpretable test samples, significantly enhancing the diagnostic capability for over-refusal. Compared to existing benchmarks, the test sets generated by DDOR have higher precision, model-specific relevance, and strong scalability.

**Strengths:**

1. Provides a valuable solution to the current challenge of "over-refusal" by creatively adapting the `ddmin` method from software testing into a tool for extracting **minimal refusal-triggering fragments (mRTFs)** in language models, enabling automatic and precise identification of the smallest refusal-inducing phrase.
2. Highly practical: does not rely on internal model parameters, making it applicable to closed-source models or rule-based safety guardrails.
3. For safe prompts that are wrongly refused, rewrites only the mRTF while preserving other parts — reducing over-refusal while maintaining the original semantics. Demonstrated effective repair of over-refusal cases on OR-Bench and XSTest, with higher average semantic similarity than full-prompt rewriting.

**Weaknesses:**

1. The overall writing and presentation quality of the paper could be further improved.
2. Trigger words in experiments (e.g., “illegal activity”) may be recognized by existing filters, potentially diminishing the repair effectiveness in real-world scenarios.

**Questions:**

1. The algorithm prunes at sentence → word granularity but does not consider semantic units (e.g., the phrase “kill a process” should remain intact). If the model is not sensitive to “kill” alone but is sensitive to the combination “kill + process,” `ddmin` may incorrectly treat the entire phrase as an mRTF or miss the actual trigger. Additionally, refusal detection relies on keyword matching (e.g., “I can’t help”), which may fail to catch indirect refusals or partial assistance.
2. Localized rewriting preserves semantics but may alter the user's true intent. For example, replacing “hack” with “access” in “how to hack a Wi-Fi” keeps surface similarity but changes the technical meaning from “illegal intrusion” to “legal connection,” essentially changing the task’s nature. The paper does not discuss how such semantic drift could impact downstream applications.

Overall, I believe addressing this challenge is inherently valuable. If the authors can improve the quality of the paper and resolve key issues, I am willing to raise my score.

---

### Note · Authors · 2026-01-23

**Comment:**

We would like to withdraw this submission. Thank you for your time and consideration.

**Withdrawal Confirmation:**

I have read and agree with the venue's withdrawal policy on behalf of myself and my co-authors.